# Integrating community health workers into the formal health system to improve performance: a qualitative study on the role of on-site supervision in the South African programme

Yu-hwei Tseng,[1] Frances Griffiths,[1,2] Julia de Kadt,[1] Nonhlanhla Nxumalo,[1] Teurai Rwafa,[1] Hlologelo Malatji,[1] Jane Goudge[1]

¹Centre for Health Policy, School of Public Health, Faculty of Health Sciences, University of the Witwatersrand, Johannesburg, South Africa
²Warwick Medical School, University of Warwick, Coventry, UK

**Correspondence to**
Dr Jane Goudge;
Jane.goudge@gmail.com

## ABSTRACT

**Objectives** To explore the role of on-site supervision in community health worker (CHW) programmes and CHW integration into the health system. We compared the functioning of CHW teams reporting to a clinic-based nurse with teams supervised by a community-based nurse. We also consider whether a junior nurse can provide adequate supervision, given the shortage of senior nurses.

**Design** A case study approach to study six CHW teams with different configurations of supervision and location. We used a range of qualitative methods: observation of CHW and their supervisors (126 days), focus group discussions (12) and interviews (117).

**Setting** South Africa where a national CHW programme is being implemented with on-site supervision.

**Participants** CHWs, their supervisors, clinic managers and staff, district managers, key informants from the community and CHW clients.

**Results** Effective supervisors supported CHWs through household visits, on-the-job training, debriefing, reviewing CHWs' daily logs and assistance with compiling reports. CHWs led by senior nurses were motivated and performed a greater range of tasks; junior nurses in these teams could better fulfil their role. Clinic-based teams with senior supervisors were better integrated and more able to ensure continuity of care. In contrast, teams with only junior supervisors, or based in the community, had less engagement with clinic staff, and were less able to ensure necessary care for patients, resulting in lower levels of trust from clients.

**Conclusion** Senior supervisors raised CHW skills, and successfully negotiated a place for CHWs in the health system. Collaboration with clinic staff reduced CHWs' marginalisation and increased motivation. Despite being clinic-based, teams without senior supervisors had lower skill levels and were less integrated into the health system.

## INTRODUCTION

Community health workers (CHWs) play an important role in improving health-care coverage and access in low-income and middle-income countries (LMICs).[1–3]

## Strengths and limitations of this study

► To the best of our knowledge, this is the first study to examine the role of different levels of supervision and location of the community health worker (CHW) team on the team's motivation, performance and integration into the health system.
► The case study approach allowed us to purposively select teams based on their supervision/location configurations, and to compare and contrast their functioning to understand what worked, and what did not.
► We employed multiple qualitative methods, including participant observation, interview and focus group discussion, to collect a substantial amount of data, enabling triangulation across data sources and sites.
► Limitations to the study include the observer effect (Hawthorne effect) during data collection, subjective rather than objective evaluations of CHW performance and data collection limited to a single health district.

However, optimising the contribution of CHWs continues to be a global challenge. CHWs, with no professional qualification, perform primary healthcare (PHC)-related functions in the community.[1 4] They work on the periphery of the health system and often in underserved communities where needs are immense and multifaceted. For this reason, CHWs require frequent and supportive supervision that supplements their training and connects them with the health system.[3 5 6] Recent reviews suggest that adequate supervision, and integration into the health system, are essential components to programme effectiveness.[5 7–9]

Effective integration of CHW programmes into the health system requires government financing, national level planning, training

and CHW scope of work, supervision, referral networks and supply chains to be connected to and incorporated into similar processes provided for other cadres.[9] Without integration, there is unlikely to be a smooth patient pathway from CHW to clinic, limiting the effectiveness of CHWs. There is a growing literature on supervision, with some evidence that supportive supervision improves motivation and performance.[10–14] However, much of this evidence is from non-governmental organisation (NGO) programmes (which are less integrated into the health system), or from programmes in which the CHWs' work is focused on a specific disease (eg, HIV) or population (eg, pregnant women). With the goal of universal access to comprehensive, integrated care, it is important to understand the type and level of supervision that will enable CHWs to provide the full range of required tasks, as well as facilitate integration into the health system.[12–15]

Given the double burden of non-communicable and infectious diseases in LMICs, governments such as South Africa's are seeking to shift CHW initiatives from focusing on a single disease or population group to a more comprehensive approach.[16] South Africa is implementing a national CHW programme (ward-based outreach teams [WBOTs]) as part of efforts to strengthen primary healthcare.[17] The intention is to provide full-time on-site supervisors whose responsibility is to manage, train, mentor and monitor CHWs and to facilitate links with the health system and the community. However, given the shortage of healthcare workers, as in other LMIC settings, there is a limited number of nurses available to supervise CHWs. In this study, we ask whether CHW teams should be linked to a clinic, and report to a clinic-based nurse who may facilitate referrals, access to supplies and CHWs' integration into the health system. Or should CHWs be based in the community in which they are working, with a dedicated nurse who does not have the additional demands of clinic duties? We also ask whether a junior (enrolled) nurse is able to provide adequate supervision, given the shortage of senior (professional) nurses. By asking these questions, we examine supervision practices, the team's integration into the health system and their effect on CHW motivation and performance.

## METHODS

In the initial observation phase (September 2016–February 2017) of a 3-year intervention study in Sedibeng Health District, Gauteng Province, we used a case study approach to examine the operation of CHW teams with different configurations of supervisors and locations: (1) clinic-based teams supervised by a senior nurse (professional nurse [PN]) and a junior nurse (enrolled nurse [EN]); (2) community (health post)-based teams supervised by a PN and an EN and (3) clinic-based teams supervised by an EN only. We studied six teams (two of each of the three configurations), each team and their supervisors being a single case study.

## Study site

In some South African provinces, CHWs, previously employed by NGOs, have been absorbed into a government WBOT programme in an attempt to shift from a patchwork of condition-specific services to a comprehensive programme with a national scope.[15 18] The CHWs' role is predominantly health promotion, prevention and screening. Standardised CHW training covers identification of the need for antenatal and postnatal care, monitoring immunisation of under 5s, adherence among patients with chronic diseases, screening for malnutrition, tuberculosis (TB), gender-based violence and making referrals to health, social and other services. WBOT teams are meant to comprise a PN, six or more CHWs, one health promoter and one environmental officer,[17] although realities on the ground often do not allow this.

We selected Sedibeng Health District as our study site due to the variation in supervision structures and location of its CHW teams. The district is relatively affluent by South African standards, although over 20% of its residents fall below the food poverty line. Outside the urban areas, disadvantaged communities with inadequate shelter, food insecurity and high disease burdens have limited access to services such as clinics, transport, water and electricity.

Sedibeng District has established 16 health posts to provide community-orientated care, different from the disease-focused care provided by 28 primary healthcare clinics.[19] A health post consists of one or two temporary structures (providing 3–6 rooms), often operating without electricity and with irregular water supply. It is managed by one or two PNs, who obtain medication and other resources via a 'mother' clinic. The nurses supervise the CHW team, and provide basic services such as chronic medication, immunisation and treating minor ailments. Some health posts are placed at some distance from a clinic, in order to improve access to basic services.

At the time of the study, there were 39 CHW teams (each with between 6 and 20 CHWs) in 37 of the district's 72 wards (the smallest geopolitical area). Sixteen of the teams were based at health posts and the remaining 23 were clinic-based. In addition to the services outlined in the national training programme (see above), the CHWs delivered medication to elderly or disabled chronically ill patients.

## Selection of CHW teams for study

The research team consulted with district officials, categorising the CHW teams into three types. We selected two teams of each type with the requirement that each pair of teams needed to be as similar as possible (important characteristics included urban or rural location, the size of the teams and the type of community they served). (We needed to pool the data from the two teams to generate a sufficient sample for the analysis of the coverage data, reported elsewhere.) The key characteristics and description of catchment areas of each team are shown in table 1. The two health post teams were closest to the community,

**Table 1** Supervision configurations of CHW teams and their catchment areas

| Type | Supervisor | Based in | Site code | Description of urban/rural location and type of community served | Maximum distance (km) CHW base to furthest household |
|---|---|---|---|---|---|
| 1 | Professional and enrolled nurse | Clinic | 1A | Township* with relatively well-off households and apartheid-era housing | 2 |
| | | | 1B | Township with RDP† housing and informal settlement‡ | 3 |
| 2 | Professional and enrolled nurse | Health post | 2A | Township | 1 |
| | | | 2B | Large informal settlement, with many migrants | 1 |
| 3 | Enrolled nurse only | Clinic | 3A | Rural, informal settlement, also with RDP housing and farm plots | 6 |
| | | | 3B | Rural, township and farm plots | 25 |

*Township: densely populated urban area built on the peripheries of towns and cities.
†RDP housing: low-cost government built housing.
‡Informal settlements: where impoverished population build shacks on vacant land.
CHW, community health worker.

while the rural teams had the largest geographical area to cover.

### Data collection

Eight fieldworkers received 2 weeks of training, covering research methods and ethics, study tools, extensive role-play and observation practice, as well as an overview of training that CHWs receive. Between September 2016 and February 2017, the field team collected interview and observation data on each team's activities, resources, engagement with the clinic and community, successes and challenges as well as client and key informant views of the performance of the teams. (Further details of CHW performance using more objective indicators, such as coverage and the quality of care, will be reported elsewhere.) Data collection method and types of data are summarised in table 2.

Fieldworkers observed CHWs and their supervisors throughout the workday, and wrote detailed daily field notes guided by a template. After each day spent in the field, the fieldworkers spent a day in the office typing up field notes in full. Each CHW or supervisor was observed for 3–5 consecutive days (interrupted by the office days) to allow for reduction in the Hawthorne effect. Participants were randomly selected, although changes were

**Table 2** Data collection method, participants and data collected

| Method | Participants | Total number in six sites | Data collected |
|---|---|---|---|
| Observation | CHWs and supervisors while conducting their daily work | 126 days of observation | Descriptions of activities, interactions and clients. |
| FGD | CHW teams | 12 FGDs (76 participants) | FGD: descriptions of activities and weekly and daily routines; resources available and needed; support from supervisors, peers, clinic and community; employment conditions; challenges of the programme. Self-administered questionnaire: age, years of training and service. |
| Interviews | Supervisors and facility managers | 43 key informant interviews | Background, training responsibilities, weekly and daily pattern of CHWs; resources; successes and challenges. |
| | Community representatives | | Perceptions of the programme; acceptability to the community needs. |
| | Follow-up interviews with CHWs' clients who were referred to the clinic during observations of household visits | 74 household interviews | Client's perception of the service, and events subsequent to the referral. |

CHW, community health worker; FGD, focus group discussion.

sometimes required due to CHWs taking leave or being absent. Participants responded well to observation, and relaxed substantially during the course of the first day of observation, although there is some evidence that the CHWs conducted more household visits early on in the observations.

All available CHWs participated in the focus group discussions (FGDs), and all supervisors and facility managers (except one) were interviewed. Community representatives were purposively sampled. Key informant interviews were recorded and transcribed. Finally, 74 household members, who received referral advice from CHWs during observed home visits, were interviewed a month later, to understand their experience of the CHW service, and whether referral advice was acted on. Interviews were conducted in participant's home, in the participant's choice of language(s) and recorded with a digital recorder. Field workers used the recording to draft a summary of the interview in English.

### Data analysis

Taking a case study approach, we drew data from the various sources to develop an explanatory description of each team, and then drew comparisons across teams. This process involved several steps. First, we extracted data from each interview or day's observation into a template, either by summarising descriptions of events, or extracting raw data such as useful quotations. This increased our familiarity with the data and allowed us to reduce its volume significantly. Multiple team members extracted data from the same sources, compared extracted data and modified extraction strategies until we were confident about inter-extractor reliability.

Second, the authors presented a brief summary of each site in a 1-day workshop. Through this collective process we identified themes that revealed the similarities and differences across the sites, such as weekly and daily pattern of activities, resources available, record keeping, managing patient referrals to the clinic, engagement with clinic staff, relationship between CHWs and supervisors, relationship with patients, local NGOs and key community stakeholders. We then generated a template into which we collated site-specific data under these themes, producing a summary for each site (see online supplementary appendix).

In preparing the manuscript, we continued to revisit the raw and summarised data to confirm descriptions of the context, enrich the content, provide clarity and check emergent ideas. The multiple data sources, as well as events documented by multiple field workers, allowed triangulation that increased the validity of our findings.

### Ethics

All participants gave informed consent. When accompanying CHWs into the community, fieldworkers obtained verbal consent from household members, prior to entering any household. There were few refusals.

### Patient and public involvement

Patients were not involved other than as respondents. Feedback has been given to provincial, district and facility management, as well as CHW supervisors and CHWs in the intervention sites in various forms, including facility level workshops.

## RESULTS

### CHW conditions of work across districts

The lack of integration led to work conditions that were demotivating for CHWs. They were not formally employed by the government, but were an outsourced workforce, managed by a private payroll company and paid a minimal stipend for 6 hours work a day. They struggled to contact the private payroll company responsible for monitoring attendance and paying stipends. An electronic system requiring CHWs to clock in/out at the clinic limited their ability to reach outlying areas as they needed to travel between the clinic and these areas on foot. Two months prior to the start of our fieldwork, CHWs had been on strike over their conditions of employment. Moreover, opportunities for career development into nursing were limited, particularly for those without a high school diploma.

CHW work was hampered by insufficient provision of logistical support due to the lack of integration. They often worked without necessary resources, including equipment, stationery, uniform, name badges or funds for transport or communication. For example, a team of 20 CHWs shared one or two blood pressure machines borrowed from the facility. When equipment, such as a glucometer, was provided, the associated consumables such as glucose strips were often not replenished. Notebooks used by CHWs to record daily activities were purchased out of their own stipend. The general lack of resources compromised CHWs' work and generated resentment.

While teams based in health posts had some dedicated space, clinic-based teams did not have sufficient space to meet, complete paper work or store their files; often files were kept at home. Supervisors did try to address these deficiencies, and some of the endeavours appeared to be morale boosting. CHWs reported that when one supervisor negotiated with the facility manager to use a meeting room for CHW meetings, it immediately increased their job satisfaction.

*Before she (PN) came, we were like orphans, we had no space inside the clinic. We were doing everything outside. She came with a lot of changes around here…PN had played an important role in our lives…I even enjoy my work now. (CHW-FGD, 1A)*

### The supervisors

The supervisors in our study sites were either senior (ie, PN) or junior (ie, EN) nurses. A PN, with a 4-year degree in nursing, is able to diagnose patients, prescribe

**Table 3** Supervision of CHW—who, what and when

| Supervision and location of teams | PN/EN clinic-based | | PN/EN health post-based | | EN clinic-based | |
|---|---|---|---|---|---|---|
| | 1A | 1B | 2A | 2B | 3A | 3B |
| Who supervises and is supervised | | | | | | |
| PN supervisor | | | | | | |
| Age (years) | 65 | 72 | 66 | 59 | n.a. | n.a. |
| Years as nurse | 36 | >30 | 38 | 39 | n.a. | n.a. |
| Years in programme | 4 | 4 | 4 | 4 | n.a. | n.a. |
| EN supervisor | | | | | | |
| Number of JN in team | 2 | 2 | 1 | 1 | 1 | 1 |
| Mean age (years) | 43 | 29 | 28 | 25 | 36 | 31 |
| Mean years as nurse | 14 | 2.5 | 0.5 | 3 | 5 | 2 |
| Years in programme | 0.6 | 0.6 | 0.6 | 0.6 | 0.3 | 0.3 |
| CHW (supervisee) | | | | | | |
| Number of CHW in team | 16 | 17 | 9 | 12 | 14 | 20 |
| Mean age in years (range) | 38 (26–53) | 39 (25–47) | 34 (27–45) | 37 (26–51) | 42 (23–58) | 33 (23–54) |
| Mean years (range) as CHW | 7 (4–12) | 7 (3–16) | 7 (6–9) | 6 (3–12) | 10 (3–9) | 6 (5–17) |
| Formal training of CHWs | | | | | | |
| % CHW received phase I training | 91% | 93% | 100% | 90% | 5.6% | 16.7% |
| % CHW received phase II training | 91% | 21% | 100% | 46% | 0% | 0% |
| Supervision activities | | | | | | |
| Supervised home visits (day/week) | | | | | | |
| By professional nurse | n.o. | n.o. | 1 | Occasional | n.a. | n.a. |
| By enrolled nurse | 3 | 4–5 | 4–5 | s.s. | 4 | On request |
| On-the-job training | Yes | Yes | Yes | Yes | n.o. | s.s. |
| Regular debriefing | Daily | n.o. | Daily | Weekly | s.s. | s.s. |
| Examining daily logs and registers | Yes | Yes | n.o. | Yes | n.o. | n.o. |
| Assisting in reporting | Yes | Yes | n.o. | Yes | n.o. | n.o. |
| Frequency of preparing reports | Weekly | Monthly | Monthly | Weekly | Weekly | Monthly |
| Resolving administrative matters | Yes | Yes | Yes | n.o. | n.o. | n.o. |

CHW, community health worker; n.a., not applicable; n.o., not observed and not indicated in other information sources; s.s., the activity was mentioned by a single source other than CHWs, but was not indicated in other data sources.

treatment and dispense medication. Importantly, the PN supervisors in the study site were trained in primary healthcare and community nursing, and had attended various other courses on TB, HIV, diabetes and hypertension, integrated management of childhood illnesses (IMCI), nursing management and leadership. Many were rehired retirees (Rehired retirees were used for three reasons. First, due to the shortage of professional nurses in the system, drawing in those who had retired increased the supply. Second, they had a wealth of experience in community nursing, which younger nurses often did not. Third, they were given an increment on top of their pension, rather than a full salary, and so cost the district less than a formally employed PN.), who had >30 years of experience as a nurse before they joined the programme (table 3). ENs have completed a 2-year course and can provide nursing care under supervision of a PN. All ENs

except one (site 3B) received 1-week induction training on the WBOT programme before joining. Some of the EN supervisors in the site had also attended courses on early childhood development, TB, prevention of mother to child transmission and the expanded immunisation programme. Most of the ENs were younger and less experienced in community work than the CHWs they supervised. The PN supervisors reported to the facility manager, as did the EN supervisors, if there was no PN supervisor in the team.

### Supervisor activities
#### Formal and informal (on-the-job) training
In four of the six sites, 90%~100% of the CHWs had completed the first phase of the national training programme, but only three sites (all with PN supervisors) had a significant number of CHWs who had passed the

second phase of training (table 3). Both phases require a written test, followed by a practical assessment conducted by the supervisor during household visits; success in this assessment requires a competent supervisor, and comprehensive and ongoing training. One PN placed considerable emphasis on training. *'When PN came, she checked who had done what course and organised training for those who needed it'* *(CHW-FGD, 1A)*. This PN also used daily morning meetings as training sessions, *'our PN is teaching us … every day in the morning'* *(CHW-FGD, 1A)*. Given the volume of information that CHWs were required to retain, these regular sessions were important. This PN involved the two ENs in the training: *'Sister teaches us like nobody's business' (EN, 1A)*, enabling the ENs to take up the supervisory role.

In contrast, in the EN-only sites, supervisors were unable to arrange formal training for the CHWs, in part due to their rural location at some distance from the district training centre, and because the ENs were not sufficiently skilled to carry out the practical assessment. With little experience themselves, they were also unable to provide on-the-job training.

### Supervised home visits

ENs routinely accompanied different pairs of CHWs on home visits several days a week. Some ENs demonstrated sensitivity in correcting CHWs' practice. *"When I can see this one didn't do it right, I keep quiet in the house, but immediately after we step outside as we are walking I do on-the-spot training" (EN, 1B)*. Other ENs working without a PN supervisor, seemed uninvolved during the home visits. Despite one EN spending 4 days a week out in the community, the CHWs did not acknowledge her as a supervisor, *"When we get there she's doing the same job that I normally do when I am alone" (CHW-FGD, 3A)*. Consequently, her CHWs did not report problems they encountered to her, such as patients with suspected TB, as they had little trust in her problem-solving capability.

### Regular debriefing

One health post-based PN/EN team (2A) held daily debriefing sessions, at which CHWs described the households visited, problems encountered and actions taken. The sessions strengthened CHWs' knowledge base and problem-solving abilities, and provided a platform for collective supervision, enabling the CHWs to learn from each other's experience.

One team (1A) held debriefings in conjunction with other activities, such as in-service training or medication preparation, which diluted the effects of debriefing. In other teams, feedback was said to happen once a month, but we were unable to validate this, and difficulties with recall of events for a whole month would likely limit the utility of this approach. Those teams that had no place to meet inside the facility tended not to discuss serious matters for reasons of confidentiality. *"Sometimes we stand outside to hold meetings… patients and the security personnel will be listening to what we are discussing" (CHW-FGD, 3A)*.

The CHWs used notebooks to document their daily activities. Through reviewing these notes, experienced supervisors could discuss the visits with the CHWs, ask questions and provide support, as observed in teams 1A, 1B and 2B. In every team there were CHWs that took poor notes, but this was better contained by supervisors willing to spend time reviewing records, and discussing problems with the relevant CHWs. It sent a clear message that documentation of activities was central to the work.

### Assistance in compiling reports

PNs of two teams (1A and 2B) required CHWs to compile their statistics every Friday, as doing it on a weekly basis (rather than monthly) reduced recall error and thus enhanced quality. One PN verified the weekly report of each CHW against their notebooks, correcting mistakes and making sure that the CHWs understood the meaning of each data category and how notes were translated into figures. This practice strengthened the ability of CHWs to record their work, and trained them to be accountable for what they did. To make sure that CHWs reported faithfully, another supervisor would make unsolicited household visits, *"Sometimes I would just go out, and when they talk about households, I would say, oh that one who lives in that shack? And they would say, 'Sister you went there?' I would say, Yes, it is necessary" (SN, 2B)*. In contrast, EN supervisors without senior support, tended to simply accept whatever reports the CHWs handed over, with no discussion or interrogation of the statistics produced.

### Clinic duties

There was an understanding between district officials and facility staff that clinic-based supervisors were to spend 70% of their time on CHW duties, and assist clinic staff in the remaining 30% of their time, particularly if the clinic was short staffed. District officials regularly restated this 'rule', as they were concerned about supervisors neglecting their CHW duties. We did observe EN supervisors, who did not have PN supervisors, spending substantial amount of time in the clinic, but this was because they felt unable to contribute to CHW activities. However, one EN with PN supervisor said:

> *if I am not busy I help with vital signs stations… it is not part of my job but I do it anyway … I refer my patients and the nurses see them. So there is teamwork. The support system is great so I never felt pressure to leave my WBOT duties. (EN interviews, 1B)*

> *Remember, (patients) in this family planning (unit of the clinic), are also the children of the mothers and the fathers that we deliver the medication at home. So, we're working hand in hand…even if they (clinic staff) ask, are you finished what you were doing? If you don't mind, can you please help me to take a blood pressure on this patient? Then I will do. (EN interview, 1A)*

### The consequences of supervision
#### Motivation, job satisfaction and engagement

Job satisfaction, professional confidence and motivation were more evident in teams with effective supervision. One CHW commented that since the PN arrived: "*we started to feel that we are now CHWs and human beings*" *(CHW-FGD, 1A)*. Two ENs supervised and mentored by PNs, explained: "*At first it was challenging for me leaving the hospital and working in the community… maybe I'm not a nurse anymore, but now I feel more useful. The community is aware of us … they are coming to us. People are taking treatment now. I think there is lesser tracing on ARV patients, most defaulters you will find are TB. Our stats are going up*" *(EN, 1B)*. "*I'm a street nurse…but you know what, when I reach the households, this is very enjoyable*" *(EN, 1A)*. When CHWs were on strike over issues with their contract in 2016, some CHWs at the clinic-based PN/EN site (1B) continued working, not wanting to let their patients down or damage their relationship with supervisors and the facility. Their supervisor interpreted their actions: "*they see that you, as their leader, are supporting them and trying to understand where they come from, especially with this employer thing being mixed up. That is why I am saying, hence, they refused to go on strike today*" *(EN, 1B)*.

However, in the absence of supportive supervision, insufficient training constrained CHWs ability to assist patients, and as a result, demotivated them. Their frustrations over their working conditions were unresolved, and sometimes unacknowledged by their immediate supervisors. Some CHWs decided to work fewer hours a day as a passive protest, in addition to participating in the collectively organised strikes.

#### Performance

CHWs with engaged and skilled supervisors carried out a fuller range of activities, and used their discretion to make decisions. For example: "*A patient had an elevated blood glucose level. However, it was too late for the patient to go to the clinic, so rather than writing a referral letter, she visited the patient again with glucometer after work, by which time the glucose level had dropped by 50%. The next morning, she went for a third time and found that sugar level had returned to normal*" *(Observation notes, 1A)*. This CHW, from a clinic-based PN/EN team, was able to interpret the glucometer readings, and rechecked the patient to avoid making an unnecessary referral.

The dedication of some CHWs to their work is evident in the following client interview: "*If you send a message to a CHW, it does not matter what time it is, or what the problem is, they will come and help you. Whatever problem you have, you can talk to them, and you are not worried that they will be talking about your problem around the area. I like the work they do very much*" *(Patient-BT, 2A)*. In the EN-only model where supervision was the weakest and support from facility staff was negligible, CHW activities were minimal, with CHWs often only delivering medication to the elderly. One CHW highlighted how a colleague's poor performance impacted on the collective reputation of the team, '*It's not good if the patient comes here and when one of the clinic staff checks the card finds out the medication delivery is long overdue. It affects all of us in the eyes of the clinic staff and the PN*' *(CHW-FGD, 1A)*.

#### Collaboration between CHWs and facility staff

Located at a distance from the main clinics, health post teams received little supervisory support from clinic staff. Other than supplying medication, there was little managerial oversight by the clinic. "*I am not directly supervising them except for sometimes when they are there. You can only supervise something that you can see. With the CHW team it is just via the phone and then it ends there*" *(Facility Manager, 2A)*. In the health post located in a community without a clinic, the PN spent a considerable amount of time seeing patients, and so neglected her CHW supervision responsibilities. Furthermore, the distance between the health post and the clinic hampered efforts to ensure CHWs' patients accessed to care at the clinic.

> *The saddest part is when the patient comes to the clinic and the file is not there, the doctor does not see the patient, and the patient is getting upset with us. Every time when we go to the clinic, we have to look for the files, the clerks do not assist us and we don't even find those files. (Observation notes from conversation with CHW, 2A)*

In clinic-based team, without a senior supervisor to assist, management of files was also a problem:

> *Sometimes you find the admin clerk is in a bad mood, she speaks to us in a bad way in front of the patients, 'I am not going to give you the file. You have to go back to the patient to tell the patient to come to the clinic to collect it'. You report to the sister, she will tell you that she has been warning (the admin clerk) for several times but she does not listen. (CHW-FGD, 3A)*

CHWs reported that other facility staff expressed a dismissive attitude towards them:

> *the peer educators, HIV/AIDS counsellor we all go together to sign the same contract but they are treated as if they are more educated than us, they call us street maids (CHW-FGD, 3B)*

When asked why they were not allowed to work inside the facility, the CHWs responded: '*the manager tells us that we are not part of the clinic, so there's nothing she can do for us*' *(field notes, 3A)*. The denial of sharing space exacerbated the CHWs' sense of marginalisation and exclusion.

However, in clinic-based teams, with the support of senior supervisors, and the resulting growth in skills and confidence, CHWs had been able to establish better relationships with clinic staff. In turn, clinic staff, seeing the CHWs' greater motivation and effort, were more likely to treat the CHWs with respect. There were several benefits to this resulting collaboration. First, communication and coordination was better. Working together, the facility manager and PN were able to confirm whether, once defaulters had been traced by CHWs, they returned to the clinic. '*We engage with facility manager, she helps with*

---

### Box 1  Support from the clinic facilitates community health worker (CHW) performance and client trust

**Case 1: Patient defaults as nurse insists on waiting for test results**

The patient, a woman on antiretroviral (ARV) drugs with a spinal cord problem, had moved recently. She was no longer able to collect ARVs and go for physio appointment at the clinic near her previous home as she had no money for transport. She asked for a referral to a closer clinic. The nurse refused as ARV patients tended to move clinics, so patient stole the file.

The CHW took her file to the closer clinic to ask if she could collect medication for the patient. The CHW was told that a new file had to be opened and the patient had to wait for CD4 results to come back. As a result, when we interviewed her, the woman had been defaulting for the past 3 weeks.

In the interview the patient said the CHW had not followed up on her requests for food parcels as they had no food, for a physio appointment, or for her husband to be tested for HIV. *"The CHW doesn't follow-up on me to see how I am doing; the only thing that she is good at is making empty promises"* (Patient-AA, 3A).

**Case 2: Social worker responds to CHW request**

The patient was a woman whose child was knocked down and killed by a car in front of her and her children. The family were traumatised. CHW wrote a referral letter to a social worker. The patient went to the social worker and then started attending a support group. CHW also arranged for the social worker to see her husband and children at home.

Patient said she trusted CHW because '*she is kind, open and she knows how to speak to people even if you are in a difficult situation. She does not gossip about her patients on the street*' (Patient-Z, 3A).

---

problems that we have…. we include her in everything' (EN, 1B). Clinic staff willingly accepted referrals from CHWs, and included the CHWs in training sessions: *'Sister X is not (in the CHW team), but she is very helpful. She attended severe malnutrition training and she called all the CHWs and did a presentation for everybody' (EN, 1B).*

Second, in one clinic, the contribution of CHWs had become indispensable: 'If they (CHWs) were not there, it means you will be over worked, the clinic will always be full, to be honest. There is good communication between us' (Facility Manager, 1A).

The third benefit was the empowerment of CHWs, who learnt to negotiate with other less willing staff: "I have a patient who doesn't want to come to the clinic and I don't want my patient to default on her HIV treatment. I said to the sister 'my patient is working at the farms so please pack medication for her. I'll ask her to come next month'. The sister scolded me because the patient did not come on her last appointment. So I begged the sister and promised to bring the patient myself. Then it was sorted, the patient got her medication and she came to the clinic with me" (CHW-FGD, 1B).

### Negotiating acceptance and building the community's trust

All CHW teams depended on the supervisor to liaise with key community members: "We established good relations with the ward councillor so whenever there was a problem I would call her" (PN, 2B). The presence of the senior supervisor increased CHWs' credibility in the community: "If CHWs found that I am not well, they report to the PN, and the PN would come and see me. I have noticed that when I follow the PN's advice, my health improves. We are alive today because of them" (Patient-BT, 2A).

Box 1 describes visits to two different households conducted on the same day by the same CHW, for which she received very different support from the facility.

Inadequate and unpredictable support from the clinic profoundly influenced the CHW's ability to provide care for her clients, and in turn, the degree of trust between client and CHW.

Table 4 summarises the key effects of location and the level of supervision on CHW performance.

**Table 4**  Impact of supervision and location on the determinants of CHW performance

| | Clinic-based with PN and EN | Health post with PN and EN | Clinic-based with EN only |
|---|---|---|---|
| Mentorship of EN and CHW | Good | Depending on how busy the PN was seeing patients. | ► Junior ENs were not able to provide adequate training or supervision.<br>► Little mentoring as facility manager was not engaged. |
| Relationship with clinic | Good | Poor | Poor |
| Files | Well managed by PN | Files at clinic were not managed by PN and EN; often lost and CHWs had to go and find them. | Part of clinic filing system. |
| Effect on quality of service | ► More able to ensure that patients get care needs<br>► Better relationship with community | ► Limited service at health post; patient had to travel to clinic for anything other than basic service.<br>► Referrals were more difficult because of distance between health post from clinic. | ► CHWs shied away from difficult cases, knowing that they did not have the support.<br>► Poorer relationship with community. |

CHW, community health worker; EN, enrolled nurse; PN, professional nurse.

## DISCUSSION

In this paper, we explored the effect of three different supervision and location configurations on CHW motivation and performance. We have demonstrated that experienced supervisors employed a range of strategies to train, motivate and monitor CHWs, to improve the quality of their work. In turn, CHWs were treated with greater respect by the clinic staff, making space for collaboration and support that benefited both parties. Being a near-equal of the facility manager, a PN supervisor was in a position to bridge the divide between the CHWs and clinic staff. When one of the PNs obtained a place for the CHWs inside the facility, she secured a position—physical and symbolic—for them in the health system. She drew in other staff to participate in the supervision of CHWs, further strengthening their legitimacy. Clinic staff provided advice and ensured that CHWs' patients accessed the care that they needed. CHWs traced the defaulting patients for the clinics and so improved the clinic's performance. The senior supervisor, a shared resource to both the CHWs and the clinic, was able to effectively integrate the CHWs into the health system.

In contrast, the teams in health posts found it difficult to build relationships with the clinic, particularly if the health post was at some distance from the clinic and the PN was busy with patient consultations. The EN-headed teams lacked adequate training and supervision, and consequently the CHWs' performance was minimal. With little evidence of the CHWs' contribution, the social distance between the CHWs and clinic staff remained substantial. Given her junior position, the EN was not able to facilitate collaboration with the clinic to reduce this distance. Hence, despite being based in the clinic, the EN-led CHW teams were not effectively integrated into the health system.

Our study has several limitations. First, despite fieldworkers asking CHWs to carry out their duties as normal, during observation many CHWs appeared to make extra effort to reach a greater number of households. We followed the same CHWs for 3–5 days to mitigate this, and the number of household visits did fall overtime, as the CHWs became used to the presence of an observer. Second, our data on CHW performance were based on what we learnt from our respondents rather than on

**Table 5** Programme design features and supervisory strategies that worked

| Recommendation | | Rationale |
|---|---|---|
| Programme design | ▶ Attachment to primary healthcare clinic | Facilitate physical and operational integration into the health system. |
| | ▶ Team up senior and junior supervisors | Build relations, guide community health workers (CHWs) to navigate through the community and system, pass down know-how to junior supervisors. |
| | ▶ Setting guidelines as to how much time supervisors can spend assisting in the clinic | Help to acknowledge that the engagement is a two-way collaboration, in which there are benefits for everybody. |
| | ▶ Strengthen HR management practices | Build trust, improve dialogue in the workplace, problem solving, supervision and culturally appropriate communication. |
| Supervisory strategies | ▶ Supervise home visits | Provide opportunities to strengthen CHWs' knowledge and skills, demonstrate a strong backup for CHWs in the community, keep updated of community's status. |
| | ▶ Formal and on-the-job training | Impart knowledge and skills. |
| | ▶ Regular debriefing/feedback | Individual and collective supervision, track performance, build up teamwork spirit. |
| | ▶ Examine daily logs and registers | Ensure accurate documentation and subsequent reporting. |
| | ▶ Direct CHWs to tasks, such as tracing defaulting patients, that explicitly assist the clinic | Ensure the clinic staff are able to see the benefits of having CHWs as part of the team. |
| | ▶ Draw in clinic staff to work with and train CHWs | Improve the extent and quality of working relationships between clinic staff and CHWs, and allow CHWs to benefit from the clinic staff's expertise. |
| | ▶ Assist data collating and reporting | Ensure that CHW activities are accurately reported so that health system managers can see the benefits of the programme. |
| | ▶ Administration and logistics | Resolve administrative matters, negotiate for better work conditions, ensure CHWs are adequately equipped to deliver service. |

an objective measure of performance. Third, while we selected study sites to ensure that the different supervisory configurations were included, we do not know to what extent the included CHW teams are representative of others in the same district. The study demonstrates noteworthy strengths. As far as we know, this is the first study to examine the effect of different levels of on-site supervisors as well as the location on CHW performance in a government programme. The case study approach allowed us to purposively select teams based on their different supervision/location configurations, and to compare and contrast their functioning in order to understand what worked, and what did not.

CHW supervision is often no more than periodic interactions by facility staff or officials who may not understand the context and role of CHWs. This can result in bureaucratic inspection and fault-finding, rather than problem solving and support.[6 7 20] For example, in the Ethiopian national programme, supervisors make irregular visits from either the local health centre or district office.[21] In our study, the presence of an appropriately skilled and qualified on-site supervisor enabled ongoing supportive engagement, rather than a distant supervisor who had fit supervisory visits into a busy schedule. Evidence from a rapid assessment of the South African WBOT programme in five provinces suggested that the existing facility-based staff were unable to provide effective supervision as they were already overstretched, working in under-resourced and underperforming clinics.[15] The authors concluded that CHW teams should be based in stand-alone health posts with a nurse supervisor, rather than within a clinic. Our data suggest it would be more advantageous for the nurse supervisor and the CHWs to be based at a clinic, with clear rules about how much time the supervisor can spend on clinic activities.

In the Ethiopian programme, CHWs based in health posts also struggled to make referrals and experienced dismissive attitudes from clinical staff; while where there was more regular contact, the CHWs felt valued and part of the team.[21] Studies in Uganda, Bangladesh and Bolivia have reported that contact with a higher level of the health system gave CHWs credibility with their clients.[22–24] Similar to our findings, in Mozambique supervision increased the participation of CHWs in the intervention facilities.[13]

Training and supervision of supervisors is a more neglected subject than that of CHWs.[9 25 26] This issue is beyond the scope of this paper, but in the second phase of our 3-year study, we are evaluating the effect on performance of a PN supervisor in building the capacity of the EN supervisors.

Globally, there are ongoing efforts to scale-up CHW programmes as a means to strengthen health systems and accelerate universal health coverage. These include the call for 1 million CHWs in sub-Saharan Africa,[27] making CHWs available everywhere on the African continent,[28] new funding commitments from the private sector,[29] as well as the recent WHO guideline on health policy and system support to optimise CHW programmes.[30]

The new WHO guideline states that *"planners should adopt 'whole-of-system' approaches, taking into account health system capacities and framing the CHW role vis-à-vis other health workers in order to appropriately integrate the CHW into the health system"*.[30] The guideline also indicates that while there is benefit of supportive supervision, there is a lack of granular evidence. Our study provides rich evidence on the range of supervision strategies, how the effectiveness of these varies by the level of supervisor and whether the team has its base in a facility or not. Taking a whole-system perspective has enabled us to draw conclusions about the barriers and facilitators to integrating the CHW into the system. Here we offer recommendations based on the experience of well-functioning teams to complement current global efforts (table 5).

## CONCLUSION

With high expectations of CHW programmes in improving access to care in South Africa and other similar settings, full-time, on-site supervision can make an important contribution in enabling CHWs to provide comprehensive, promotive and preventative care. In the context of an overstretched health system, senior supervision located within a clinic can enable the development of a collaboration between the clinic and CHWs, reducing the CHWs' marginalisation, and enabling their potential contribution to improving health outcomes to be realised.

**Acknowledgements** The authors would like to thank the following individuals for their invaluable contribution: the field workers, the respondents, the CHW teams, Mrs Salamina Hlahane, district manager at Sedibeng and Mrs Bridget Lefhoedi, community-based services coordinator. Input from Professors Margaret Thorogood, David Sanders and Emmanuelle Daviaud throughout the study has been highly appreciated. The authors would also like to thank these organisations for their generous support: Department of Health Sedibeng District, Gauteng Department of Health, the UK Medical Research Council (MRC) and Taiwan Ministry of Science and Technology.

**Contributors** YHT analysed and interpreted data, wrote and revised the manuscript with JG, and approved the final manuscript as submitted. FG and JG conceptualised and designed the study, analysed data, critically revised the manuscript and approved the final manuscript as submitted. JdK carried out data collection process, managed data, analysed data, critically revised the manuscript and approved the final manuscript as submitted. NN analysed data, revised the manuscript and approved the final manuscript as submitted. TR and HM analysed data and approved the final manuscript as submitted.

**Funding** MRC-UK under Health Systems Research Initiative (MR/N015908/1). Taiwan Ministry of Science and Technology (105-2917-I-564-006).

**Competing interests** None declared.

**Patient consent for publication** Obtained.

**Ethics approval** The study protocol and instruments were approved by the University of the Witwatersrand's Human Research Ethics Committee (Medical) (M160354), the Gauteng Provincial Health Research Committee in South Africa and the Biomedical and Scientific Research Ethics Committee (BSREC) at Warwick University (REGO-2016–1825), UK.

**Provenance and peer review** Not commissioned; externally peer reviewed.

**Data sharing statement** A sample of site-specific summary is available in appendix.

and indication of whether changes were made. See: https://creativecommons.org/licenses/by/4.0/.

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
