## [Reviewer comments · BMJ Open]

ARTICLE DETAILS

TITLE (PROVISIONAL)	Integrating community health workers into the formal health system to improve performance: a qualitative study on the role of on-site supervision in the South African programme
AUTHORS	Tseng, Yu-hwei; Griffiths, Frances; de Kadt, Julia; Nxumalo, Nonhlanhla; Rwafa, Teurai; Malatji, Hlologelo; Goudge, Jane

VERSION 1 – REVIEW

REVIEWER	Karen Daniels South African Medical Research Council
REVIEW RETURNED	19-Mar-2018

GENERAL COMMENTS	Manuscript ID: bmjopen-2018-022186 Thank you for the opportunity to review this manuscript. It is always important to have voices from LMICs telling the stories about issues impacting on health care within LMICs. Background: 1) While supervision was already identified in the late 1980s as a weak link in lay health worker programmes, there was a lag in focus on supervision in practice in the literature. Recently however, there has been more of a focus on this subject. This recent literature is not sufficiently reflected in the background or the discussion of this manuscript. The authors cite some of the literature, but they don't discuss it. It is important for an academic paper to show its contribution to science or to knowledge, by discussing and critiquing the contemporary and historical literature, and then to show how this new paper fills a gap, and contributes new knowledge. The authors also miss important pieces of literature. These include a systematic review by Glenton et al and a primary studies by Daniels et al, and Akintola et al (all references given below). Glenton et al, include both a critique on how supervision has been reported on in the literature, as well as an analysis of the findings on supervision from papers focused on maternal and child health. Daniels et al, focuses on the experience of supervision within a South African research project. It would be important to compare the findings from your study, in which supervision is done by nurses who have other tasks beyond the supervision of lay health workers, to that of Daniels et al, where as a research project, supervision was a dedicated, full time task. Akintola and Chikoko, explore the experiences of supervisors working in community based organisations. Again, an important comparison, where the supervisors themselves have been lay health workers, but do not have the professional training of nurses. What does it mean when one puts these three cases together – nurse supervisors in the formal health sector, supervisors who are part of a research study, and supervisors from community based
--

organization, who once themselves were lay health workers – what is the new knowledge? Please also don't stop at these studies. Use Google Scholar to see who has cited Daniels and Akintola (e.g. you'll find some studies from Mozambique), and explore your study against these studies too.

- Glenton C, Colvin CJ, Carlsen B, Swartz A, Lewin S, Noyes J, Rashidian A. Barriers and facilitators to the implementation of lay health worker programmes to improve access to maternal and child health: qualitative evidence synthesis. *Cochrane Database Syst Rev.* 2013 Oct 8;(10):CD010414. doi:b10.1002/14651858.CD010414.pub2. Review. PubMed PMID: 24101553.

- Daniels et al., Supervision of community peer counsellors for infant feeding in South Africa: an exploratory qualitative study, *Human Resources for Health* 2010, 8:6

- Akintola and Chikoko, Factors influencing motivation and job satisfaction among supervisors of community health workers in marginalized communities in South Africa, *Human Resources for Health* (2016) 14:54

2) The authors use the term community health worker, yet the persons being referred are based at formal health facilities from which they then go into the community. Please consider using the term lay health worker, as suggested by Lewin et al 2010 (whom you cite), as this is a more accurate, even if less popular descriptor.

3) The authors claim that South Africa is in the process of developing a national CHW programme. In my understanding there was period when there was a flurry of activity around this, up until around 2010, but since then this has stalled. The new activity has been around ward based out reach teams (WBOTs), which is not addressed in the manuscript.

Methods:

4) Please let the reader know upfront what the study period is, both for the larger intervention study as well as for this sub-study. One has to wait until the section on data collection to discover this.

5) Please give dates for when "Two years before the introduction of the national CHW policy" is? Also, which policy are the authors referring to? In my understanding there were two before WBOTs, and WBOTs may come with its own policy.

6) What does "Cuban-style health posts" mean? Your readers will be based all across the world, and may not know the South African context, so explain what you mean and avoid colloquialisms.

7) Which national programme took effect in 2011?

8) The last paragraph on Study site, reads like findings. Either move this to the findings or cite the source of the information that you're using to describe the site.

9) Both the data collection and data analysis sections offer inadequate description of what was done by the team. The reader needs to be convinced of the trustworthiness of these processes. In qualitative research, the researcher is the most important instrument, therefore this role must be made absolutely transparent. What role did the first author play? What did other members of the team do? Who amongst the authors collected the data? How was data collection prepared for (e.g. what training did the first author and other authors receive in data collection)? Where was the data collected (homes, facilities, quiet room, etc)? How long did the interviews and focus groups last? How did participants respond to being observed and being interviewed? How did clients react to having data collectors tag along? How were the interviews and focus groups recorded (e.g. were digital

	recorders used)? What language was used in data collection? Explain the data translation process (who and how), if any. Who transcribed the data? How was the data stored? 10) The authors claim that they did thematic content analysis, when in fact there is no indication that this was done. I don't know framework analysis well, but it reads like this was the approach taken? In following my comments above, the label for what was done is not as important as the description of what was done. The authors claim to have done 12 focus groups, plus 43 interviews, as well as 126 person days of observations. This is an enormous amount of data. How was all of it handled? Who did what? Which authors looked at site 1, which authors looked at site and 3? How were the findings from the different authors synthesized? What groupings were made? The data analysis process must be transparently and explicitly explained, because it is currently very very vague, and this vagueness raises questions about rigour and trustworthiness. 11) The manuscript says nothing about recruitment. This needs to be explained in full, including explanation as to whether or not anyone refused, and if so, why not? Was ethics consent sought from clients for the observations? 12) It is completely unclear who was included in the recruitment? Initially I thought that it was only nurses and lay health workers, but the findings have quotes from patients. Please use a table to explain who the participants were. Findings: 13) The findings are reported as if they are answers to questions. No themes are reported. This suggests that a deductive rather than an inductive approach was used in the analysis, where the authors sought to answer a set of questions, rather than look for patterns and themes. The analysis remains at a manifest, descriptive level. This is fine, but the authors need to be upfront about this. 14) The quotes included are very brief, and none of the extracts from the observations are included. The reader needs more substantial extracts from the raw data, to justify the authors interpretation of the findings. 15) Please keep quotes and data extracts on separate lines, for ease of reading, and so as to allow the reader to quickly access see and assess them. Discussion: As per the Background, this section is very thin and needs to be substantiated by situating this study within the broader literature.
--	--

REVIEWER	Abimbola Olaniran Liverpool School of Tropical Medicine, United Kingdom
REVIEW RETURNED	25-Mar-2018

GENERAL COMMENTS	The authors have identified an important topic in relation to CHW programmes. I agree with the authors that CHW supervision is one of the weakest elements in many CHW programmes. As such, this paper has the potential to address this weak link after addressing key issues relating to the manuscript: INTRODUCTION 1. Performance is a broad term and nuanced differently across stakeholder groups. Largely, many consider performance to be either (or a combination of) quality of service delivery, output or outcome of service delivery. The authors need to clarify the term performance to the readers emphasising which aspect they set out to measure.
---

	METHOD The method section would benefit from appropriate use/clarification of the various terminologies used in this section:  1. The authors have stated that a mixed method intervention was done. Readers may need additional information to differentiate this from the study method (qualitative study). 2. The authors have stated that an observation study was done. Readers may be keen to know if this is the study design or one of the data collection methods (participant observation). 3. The authors need to clarify the sampling technique used for identifying the districts and selecting the study participants. Readers would benefit from additional information on the criteria used for the purposive sampling. 4. The authors have mentioned that both deductive and inductive approaches were used to code data and identify themes. However, it is not very clear how this was done, especially the deductive approach. RESULTS AND DISCUSSION The results section would benefit from a comparison of the study findings across the 3 models to see if there are intergroup differences and the factors responsible for these differences. Identification and documentation of these factors may inform policy, practice and future research relating to supervision of CHWs. Overall, I a missing the article’s contribution to policy and practice.
--	--

REVIEWER	Delanyo Dovlo WHO Regional Office for Africa, Congo
REVIEW RETURNED	02-Apr-2018

GENERAL COMMENTS	more information is provided in the attached document. The paper is well written and covers a subject of high importance – CHWs, an area that is again reemerging in global health as an instrument towards UHC. The CHW issue has remained rather ill-defined due to the fact a CHW system has to be quite context related. Supervision is of course an essential element of whether CHWs function well or not – other important factors include training, equipment, incentives and motivators and the community acceptance (influenced by various other social and demographic characteristics. As the paper focuses on supervision in a specific location of a province in South Africa. While the study environment and context is fairly well described, it may be difficult for people unfamiliar with the health systems in South Africa or the Gauteng Province to appreciate the facilities and staff types described and these perhaps need to be more elaborated (e.g., define the difference between a health post and clinic, and between a senior nurse and junior nurses indicating a bit more on the difference in training. For example – will a junior nurse (enrolled nurse) with 20 years’ experience, have better supervisory skills than he relatively junior ones compared with retired senior nurses in the study? Further comments are in attachment The effect of full time supervisors on the performance of a South African community health worker programme: An observation study
--

	reviewer: Delanyo Dovlo The paper is well written and covers a subject of high importance – CHWs, an area that is again reemerging in global health as an instrument towards UHC. The CHW issue has remained rather ill-defined due to the fact that communities and their health issues differ widely and by nature, a CHW system has to be quite context related. Supervision is of course an essential element of whether CHWs function well or not – other important factors include training, equipment, incentives and motivators and the community acceptance (influenced by various other social and demographic characteristics). The paper focuses on supervision in a specific location of a province in South Africa. While the study environment and context is fairly well described, it may be difficult for people unfamiliar with the health systems in South Africa or the Gauteng Province to appreciate the levels of facilities involved and what types of nurses performed the supervision and these perhaps need to be elaborated on a bit more (define the difference between a health post and clinic, and between a senior nurse and junior nurses indicating a bit more on the difference in training. For example – will a junior nurse (enrolled nurse) with 20 years' experience, have better supervisory skills than he relatively junior ones compared with retired senior nurses in the study? Supervision that is effective can be quite resource intensive – time, pay, travel costs etc). The senior nurses were rehired retirees, the junior nurses seemed to be younger in-service staff – were the junior nurses also solely dedicated to CHW supervision or had other facility duties? This may need to be more clearly explained – as the comparison with senior nurses supervisory advantage may be due to other factors – time, experience, community acceptance (of young supervisors for example). The training program afforded to the community health workers was fairly well described, but I could not find the training and preparation that the supervisors, whether senior or junior, had to undertake in order to play their roles. The effectiveness of community health workers and their supervisors and indeed the support from the facilities, I believe will involve alignment of a number of “hard” and “soft” incentives and motivations – some of which came out well in the study (eg building relations and being seen to support the pay and work issues of the CHWs). However, achieving work effectiveness for both CHWs and supervisors goes beyond their interrelationship and this could have been explored a bit more. For example; What advantage does CHWs bring to the clinics and health posts' work? How does these help facility staff to do their work better? Are there motivational factors for the retired senior nurses that the junior nurses lacked beyond being younger and inexperienced? It may be good also to explain to readers why rehired retirees were used and not current facility staff The study described and discusses a number of interesting arrangements and interactions that produced a good effect (and some that did not) and does give quite a number areas to deduce ways of setting out a good supervisory mechanism for CHWs (indeed for all field workers) – about note taking, weekly statistics reviews; sampling previously visited households for the supervisor to visit; proving space to integrate CHWs work into clinic life etc etc., which could perhaps be summarized into a set of clearer
--	--

	recommendations of what worked and what didn't work. Arguably the sample and the context is limiting but can still generate enough dimensions for future reviews and studies. With Africa's high NCD burden, it is good to see the NCD component of CHW work highlighted (rather than the usual focus on the global and donor supported initiatives. Perhaps a future study could look at the role of these CHWs in the non-communicable diseases challenge in communities. I believe there should be some overarching oversight of supervisors, which has not been described in the study (who supervises them and how is that done? Does it influence their performance? Who do they report to?) What is the policy within which the CHWs function? A brief summary will help an external reader. In general, I recommend these minor revision and clarification of information to make the article even more informative and relevant.
--	--

VERSION 1 – AUTHOR RESPONSE

Reviewer: 1

Reviewer Name: Karen Daniels

Institution and Country: South African Medical Research Council

Please state any competing interests: I know some of the authors well.

Manuscript ID: bmjopen-2018-022186

Thank you for the opportunity to review this manuscript. It is always important to have voices from LMICs telling the stories about issues impacting on health care within LMICs.

Background:

1) While supervision was already identified in the late 1980s as a weak link in lay health worker programmes, there was a lag in focus on supervision in practice in the literature. Recently however, there has been more of a focus on this subject. This recent literature is not sufficiently reflected in the background or the discussion of this manuscript. The authors cite some of the literature, but they don't discuss it. It is important for an academic paper to show its contribution to science or to knowledge, by discussing and critiquing the contemporary and historical literature, and then to show how this new paper fills a gap, and contributes new knowledge. The authors also miss important pieces of literature. These include a systematic review by Glenton et al and a primary studies by Daniels et al, and Akintola et al (all references given below). Glenton et al, include both a critique on how supervision has been reported on in the literature, as well as an analysis of the findings on supervision from papers focused on maternal and child health. Daniels et al, focuses on the experience of supervision within a South African research project. It would be important to compare the findings from your study, in which supervision is done by nurses who have other tasks beyond the supervision of lay health workers, to that of Daniels et al, where as a research project, supervision was a dedicated, full time task. Akintola and Chikoko, explore the experiences of supervisors working in community based organisations. Again, an important comparison, where the supervisors themselves have been lay health workers, but do not have the professional training of nurses. What does it mean when one puts these three cases together – nurse supervisors in the formal health sector, supervisors who are part of a research study, and supervisors from community based organization, who once themselves were lay health workers – what

is the new knowledge? Please also don't stop at these studies. Use Google Scholar to see who has cited Daniels and Akintola (e.g. you'll find some studies from Mozambique), and explore your study against these studies too.

Response:

We appreciate the suggestions. The useful references provided by the reviewer facilitated a new round of informed literature review. The three studies, as well as other relevant papers have been included and discussed in both introduction and discussion. Main changes can be found on pages 4-6 lines 4-40 and pages 32-34 line 490 -511. Rather than picking up on Karen's suggestion of comparing supervision in different types of programmes, we have re-focused the paper slightly to highlight the fact that our study design allowed us to explore the difference between senior / junior supervision and how supervision, as well as location influenced how well integrated the CHWs were into the health system. In the introduction we explain the importance of integration for CHW effectiveness, and in the paper we then examine how supervision facilitated this.

2) The authors use the term community health worker, yet the persons being referred are based at formal health facilities from which they then go into the community. Please consider using the term lay health worker, as suggested by Lewin et al 2010, as this is a more accurate, even if less popular descriptor.

Response:

Definitions for lay health workers (LHWs) and community health workers (CHWs) share overlapping characteristics. Lewin and colleagues (2010) used LHWs in their seminal paper, and at the same time acknowledged a broad scope of terminology by accommodating CHWs, village health workers, treatment supporters, etc. depending on the context where the reference is situated.

While we appreciate that the LHW is a more inclusive phrase, including non-professionals who in work in the community and those who work in clinic and other settings, in this paper we would prefer to keep the term CHW as in the programmes we are describing their main tasks are out in the community. Moreover CHW currently denotes WBOT members in South Africa. CHWs are used by researchers (for example, Doherty, Tanya, Kroon, Rhoda, and Sanders. 2016; Marcus, Hugo, and Jinabhai. 2017; Trafford, Swartz, and Colvin. 2018. Secondly the term is used in official documents (for example, South Africa Department of Health, 2011, Provincial guidelines for the implementation of the three streams of PHC Re-engineering), and, finally, by CHWs themselves.

3) The authors claim that South Africa is in the process of developing a national CHW programme. In my understanding there was period when there was a flurry of activity around this, up until around 2010, but since then this has stalled. The new activity has been around ward based outreach teams (WBOTs), which is not addressed in the manuscript.

Response:

We apologize for the confusion. A clarification has been added in the **Introduction:**

“Given the double burden of non-communicable and infectious diseases in LMICs, governments such as South Africa's are seeking to shift CHW initiatives from focusing on a single disease or population group, to a more comprehensive approach.[16] South Africa is implementing a national CHW programme (WBOTs; ward-based outreach teams) under a policy to strengthen primary health care [17]. The intention is to provide full-time on-site supervisors whose responsibility is to manage, train,

mentor, monitor CHWs and to facilitate links with the health system and the community.” (page 5, lines 26-32)

and in more detail in a background subsection in Methods, under the heading of Study Site:

“In some South African provinces, CHWs, previously employed by non-governmental organizations (NGOs), have been absorbed into a government programme (called WBOTS; ward-based outreach teams) in an attempt to shift from the patchwork of services to a programme with a national scope. Standardized CHW training covers registrations of households to identify the need for ante-natal and post-natal care, monitoring adherence among patients with chronic diseases, and to the immunisation programme for under 5s, screening for malnutrition, TB, gendered based violence, and making any resulting referrals to health, social and other services. The CHW role is predominately promotion and prevention, screening and referral. WBOT teams are meant to comprise a professional nurse, six or more CHWs, one health promoter and one environmental officer.” (Page 6-7, lines 55-65)

Methods:

4) Please let the reader know upfront what the study period is, both for the larger intervention study as well as for this sub-study. One has to wait until the section on data collection to discover this.

Response:

We have added the following to the final paragraph of the introduction.

“In the initial observation phase (Sept 2016- Feb 2017) of a 3-year intervention study in Sedibeng Health District, we studied CHW teams with different configurations of supervisors and locations...”
“(page 6, Line 42)

5) Please give dates for when “Two years before the introduction of the national CHW policy” is? Also, which policy are the authors referring to? In my understanding there were two before WBOTs, and WBOTs may come with its own policy.

Response:

The vague sentence “Two years...” has been replaced with a clear-cut statement, followed by a description of pre-existing health posts in Sedibeng.

Sedibeng District has established 16 health posts to provide community-orientated care, rather than the disease focused care provided by 28 primary health care clinics.[19] Some health posts were placed at some distance from a clinic, in order to improve access to basic services. A health post consists of one or two temporary structures (providing 3-6 rooms), without electricity and often with irregular water supply. It is managed by one or two professional nurses, who obtain medication and other resources via a ‘mother’ clinic. The nurses supervise the CHW team, and provide basic services such as chronic medication, immunisation and treating minor ailments.

At the time of the study there were 39 CHW teams (each with between 6-20 CHWs) in 37 of the district's 72 wards (smallest geopolitical area). Sixteen of the teams were based at a health post and the remaining 23 were clinic-based. In addition to the services outlined in the national training programme (see above), the CHWs delivered medication to elderly or disabled chronically ill patients. (page 8, lines 83-87)

6) What does "Cuban-style health posts" mean? Your readers will be based all across the world, and may not know the South African context, so explain what you mean and avoid colloquialisms.

Response:

Please refer to response for point 5) above for a revised description of the characteristics of health posts in Sedibeng.

7) Which national programme took effect in 2011?

Response:

It is the PHC Re-engineering. The revised manuscript is clearer and reads:

"South Africa is implementing a national CHW programme (WBOTs; ward-based outreach teams) under a policy to strengthen primary health care [17]."

(lines 28-30, page 5) Reference 17 used here is the government policy statement on PHC re-engineering.

8) The last paragraph on Study site, reads like findings. Either move this to the findings or cite the source of the information that you're using to describe the site.

Response:

Thanks for pointing this out. We have moved this paragraph into the findings section. It was shifting this paragraph that led us to re-think the focus of the paper, mentioned under point 1 above.

9) Both the data collection and data analysis sections offer inadequate description of what was done by the team. The reader needs to be convinced of the trustworthiness of these processes. In qualitative research, the researcher is the most important instrument, therefore this role must be made absolutely transparent. What role did the first author play? What did other members of the team do? Who amongst the authors collected the data? How was data collection prepared for (e.g. what training did the first author and other authors receive in data collection)? Where was the data collected (homes, facilities, quiet room, etc)? How long did the interviews and focus groups last? How did participants respond to being observed and being interviewed? How did clients react to having data collectors tag along? How were the interviews and focus groups recorded (e.g. were digital

recorders used)? What language was used in data collection? Explain the data translation process (who and how), if any. Who transcribed the data? How was the data stored?

Response:

The data collection section has been greatly expanded, according to the reviewer’s suggestion. A new table is added to provide an overview of data collected.

Data collection (page 9-10, lines 105-117)

Eight fieldworkers received two weeks of training, covering research methods and ethics, study tools, extensive role-play and observation practice, as well as an overview of training that CHWs receive. Between Sept 2016 and Feb 2017, the field team collected interview and observation data on each teams’ activities, resources, engagement with the clinic and community, successes and challenges, as well as respondent’s views of the performance of the teams. (Further details of CHW performance using more objective indicators, such as coverage and the quality of care, will be reported elsewhere.) Data collection method and types of data are summarised in Table 2

Table 2: Data collection method, participants and data collected

Method	Participants	Total number in 6 sites	Data collected
Observation	CHWs and supervisors while conducting their daily work	126 days of observation	Descriptions of activities and interactions; descriptions of clients encountered;
	CHW teams	12 FGDs (76 participants)	FGD : descriptions of activities and weekly & daily routines; resources available and needed; support from supervisors, peers, clinic and community; employment conditions; challenges of the programme Self-administered questionnaire : age, years of training and service,
Interviews	Supervisors and facility managers	43 key informant interviews	Background, training responsibilities, weekly & daily pattern of CHWs; resources; successes and challenges
	Community representatives		Perceptions of the programme; acceptability to the community needs;
	Follow-up interviews with CHWs’ clients who were referred to the clinic during	74 household interviews	Client’s perception of the service, and events subsequent to the referral

Method	Participants	Total number in 6 sites	Data collected
	observations of household visits		

¹FGD: focus group discussion

Fieldworkers observed CHWs and their supervisors throughout the workday, and wrote detailed daily field notes guided by a template. After each day spent in the field, the fieldworkers spent a day in the office to type up field notes in full. Each CHW or supervisor was observed for 3-5 consecutive days (interrupted by the office days) to allow for reduction in the Hawthorne effect. Participants were randomly selected, although changes were sometimes required due to CHWs taking leave or being absent. Participants responded well to observation, and relaxed substantially during the course of the first day of observation, although there was some evidence the CHWs conducted more household visits early on in the observations.

All available CHWs participated in the FDGs, and all supervisors and facility managers (except one) were interviewed. Community representatives were purposively sampled.

Finally, 74 household members, who received referral advice from CHWs during observed home visits, were interviewed a month later, to understand their experience of the CHW service, whether referral advice was acted upon, and whether the follow up service was effective. Interviews were conducted in participant homes, in the participant's choice of language(s), and recorded with a digital recorder. Field workers used the recording to draft a summary of the interview in English.

10) The authors claim that they did thematic content analysis, when in fact there is no indication that this was done. I don't know framework analysis well, but it reads like this was the approach taken? In following my comments above, the label for what was done is not as important as the description of what was done. The authors claim to have done 12 focus groups, plus 43 interviews, as well as 126 person days of observations. This is an enormous amount of data. How was all of it handled? Who did what? Which authors looked at site 1, which authors looked at site and 3? How were the findings from the different authors synthesized? What groupings were made? The data analysis process must be transparently and explicitly explained, because it is currently very very vague, and this vagueness raises questions about rigour and trustworthiness.

Response:

The data analysis section has been re-written, according to the reviewer's suggestion. (While we have described the process in much more detail, we haven't provided explicit information about which team members did what .. just in the interest of keeping the word count down. However, if you feel we need more information on this, we can add it.)

Data analysis (page 11-12, lines 138-161)

Taking a case study approach, we drew data from the various sources to develop an explanatory description of each team, and then drew comparisons across teams. This process involved several steps. Firstly, we extracted data from each interview or day's observation into a template, either by summarising descriptions of events, or extracting raw data such as useful quotations. This increased our familiarity with the data and allowed us to reduce its volume significantly. Initially, multiple team members extracted data from same sources, compared extracted data and modified extraction strategies until we were confident about inter-extractor reliability.

Secondly, the authors presented a brief summary of each site in a one-day workshop. Through this collective process we identified themes that revealed the similarities and differences across the sites, such as weekly and daily pattern of activities, resources available, record keeping, managing patient referrals to the clinic, engagement with clinic staff, relationship between CHWs and supervisors, relationship with patients, local NGOs, and key community stakeholders. We then generated a template into which we collated site-specific data under these themes, producing a summary for each site (see example in appendix).

In preparing the manuscript, we continued to revisit the raw and summarised data to confirm descriptions of the context, enrich the content, provide clarity, and check emergent ideas, throughout data interpretation and manuscript writing. The multiple data sources, as well as events documented at times by multiple field workers, allowed triangulation that increased the validity of our findings.

11) The manuscript says nothing about recruitment. This needs to be explained in full, including explanation as to whether or not anyone refused, and if so, why not? Was ethics consent sought from clients for the observations?

Response:

For recruitment of CHW teams, we have added in lines 90-98, page 8-9:

“The research team consulted with district officials, categorizing the CHW teams into three types (See Table 1). We selected two teams of each type with the requirement that each pair of teams needed to be as similar as possible (important characteristics included urban or rural location, the size of the teams, and the type of community they served). (We needed to pool the data from the two teams to generate a sufficient sample for the analysis of the coverage data, reported elsewhere). Tables 1 and 3 describe the characteristics of different teams and their catchment areas. The two health post teams were closest to the community, while the rural teams had the largest geographical area to cover (Table 1).

In addition, the ethics section now reads:

“All participants gave informed consent. When accompanying CHWs into the community, fieldworkers obtained verbal consent from household members, prior to entering any household, when observing CHW visits. There were few refusals.” (page 13, lines 168-170)

12) It is completely unclear who was included in the recruitment? Initially I thought that it was only nurses and lay health workers, but the findings have quotes from patients. Please use a table to explain who the participants were.

Response:

This comment has been well-received and addressed in a table that summarizes the types and number of respondents, and the types of data that they provided. It is presented in the response for point 9.

Findings:

13) The findings are reported as if they are answers to questions. No themes are reported. This suggests that a deductive rather than an inductive approach was used in the analysis, where the authors sought to answer a set of questions, rather than look for patterns and themes. The analysis remains at a manifest, descriptive level. This is fine, but the authors need to be upfront about this.

Response:

Thank you for requesting that we be more specific about our approach to analysis. We used case study approach. The re-written data analysis section (provided in response to point 10 above) details how we developed explanatory descriptions of the functioning of each team, how we derived the main themes of comparison across sites, and how we then conducted that comparison. We now have written more clearly how we step by step handled data, as indicated in the response for point 10.

14) The quotes included are very brief, and none of the extracts from the observations are included. The reader needs more substantial extracts from the raw data, to justify the authors interpretation of the findings.

Response:

We have added a considerable number of quotations throughout the findings section, and have revisited the whole findings section to ensure that our interpretations are closely grounded in the quotations

15) Please keep quotes and data extracts on separate lines, for ease of reading, and so as to allow the reader to quickly access see and assess them.

Response:

Where we have included quotations 3 lines or longer we have started the quotation on a separate line. However, we are of the view that including short quotations in the text is not unusual, and helps with the flow of the paper. If the journal has a particular rule on this, we would be happy to follow it.

Discussion:

16) As per the Background, this section is very thin and needs to be substantiated by situating this study within the broader literature.

Response:

Thank you. This section has been substantially revised, on pages 30-35.

Reviewer: 2
Reviewer Name: Abimbola Olaniran
Institution and Country: Liverpool School of Tropical Medicine, United Kingdom
Please state any competing interests: No competing interest

The authors have identified an important topic in relation to CHW programmes. I agree with the authors that CHW supervision is one of the weakest elements in many CHW programmes. As such, this paper has the potential to address this weak link after addressing key issues relating to the manuscript:

INTRODUCTION

1. Performance is a broad term and nuanced differently across stakeholder groups. Largely, many consider performance to be either (or a combination of) quality of service delivery, output or outcome of service delivery. The authors need to clarify the term performance to the readers emphasising which aspect they set out to measure.

Response:

Thank you. In this paper we have chosen to focus on respondent's assessment of CHWs motivation and performance. The study has collected more objective measures of performance, such as coverage and quality of care, however, we have decided to report these in another paper. We have added the following text in the data collection section (page 9-10, lines 108-114) now reads

"Between Sept 2016 and Feb 2017, the field team collected interview and observation data on each teams' activities, resources, engagement with the clinic and community, successes and challenges, as well as respondent's views of the performance of the teams. (Further details of CHW performance using more objective indicators, such as coverage and the quality of care, will be reported elsewhere.) Data collection method and types of data are summarised in Table 2"

In the discussion section we have noted the reliance on our respondent's view of the CHW performance as a limitation: (page 31, lines 478-480)

"Second, the information on performance is based on what we learned from our respondents rather than on an objective measure of performance.

METHOD

The method section would benefit from appropriate use/clarification of the various terminologies used in this section:

1. The authors have stated that a mixed method intervention was done. Readers may need additional information to differentiate this from the study method (qualitative study).

Response:

Thank you. We have added additional information at the end of the introduction (on page 6, lines 51-52), clarifying that this paper reports on the initial observation phase of a 3-year intervention study (see response under point 1 above)

The first line of the methods section is now explicit about our study design for the first phase:

“We used a case study approach to examine the operation of six CHW teams, with each team and their supervisors as a single case study.”

2. The authors have stated that an observation study was done. Readers may be keen to know if this is the study design or one of the data collection methods (participant observation).

Response:

Thank you. We have now clarified that this paper reports on the initial observation phase of a 3 year intervention study, and that we used a case study design. (See above)

3. The authors need to clarify the sampling technique used for identifying the districts and selecting the study participants. Readers would benefit from additional information on the criteria used for the purposive sampling.

Response:

For the selection of the CHW teams we have added the following text, lines 90-98, page 8-9:

“The research team consulted with district officials, categorizing the CHW teams into three types (See Table 1). We selected two teams of each type with the requirement that each pair of teams needed to be as similar as possible (important characteristics included urban or rural location, the size of the teams, and the type of community they served). (We needed to pool the data from the two teams to generate a sufficient sample for the analysis of the coverage data, reported elsewhere). Tables 1 and 3 describe the characteristics of different teams and their catchment areas. The two health post teams were closest to the community, while the rural teams had the largest geographical area to cover (Table 1).

For the selection of CHWs to be observed, we have added the following text:

“Participants were randomly selected, although changes sometimes required due to CHWs taking leave or being absent.” (page 10, line 123-124), and

“All available CHWs participated in the FGDs, and all supervisors and facility managers (except one) were interviewed.” (page 11, line 129-130)

The key informants were purposively selected.

4. The authors have mentioned that both deductive and inductive approaches were used to code data and identify themes. However, it is not very clear how this was done, especially the deductive approach.

Response:

The data analysis section has been completely re-written to explain more clearly step by step how we handled data. Using a case study approach, we developed explanatory descriptions of the functioning of each team, then we derived the main themes of comparison across sites, and then we conducted that comparison. Our approach was therefore predominately inductive, drawing on the data, however, as we developed themes of the comparison, we did have in mind our knowledge of key issues that have been documented in the CHW literature, which, inevitably to some extent shaped what we focused on. (lines 138-161, page 11-12)

“Taking a case study approach, we drew data from the various sources to develop an explanatory description of each team, and then drew comparisons across teams. This process involved several steps. Firstly, we extracted data from each interview or day’s observation into a template, either by summarising descriptions of events, or extracting raw data such as useful quotations. This increased our familiarity with the data and allowed us to reduce its volume significantly. Initially, multiple team members extracted data from same sources, compared extracted data and modified extraction strategies until we were confident about inter-extractor reliability.

Secondly, the authors presented a brief summary of each site in a one-day workshop. Through this collective process we identified themes that revealed the similarities and differences across the sites, such as weekly and daily pattern of activities, resources available, record keeping, managing patient referrals to the clinic, engagement with clinic staff, relationship between CHWs and supervisors, relationship with patients, local NGOs, and key community stakeholders. We then generated a template into which we collated site-specific data under these themes, producing a summary for each site (see example in appendix).

In preparing the manuscript, we continued to revisit the raw and summarised data to confirm descriptions of the context, enrich the content, provide clarity, and check emergent ideas, throughout data interpretation and manuscript writing. The multiple data sources, as well as events documented at times by multiple field workers, allowed triangulation that increased the validity of our findings.”

RESULTS AND DISCUSSION

The results section would benefit from a comparison of the study findings across the 3 models to see if there are intergroup differences and the factors responsible for these differences. Identification and documentation of these factors may inform policy, practice and future research relating to supervision of CHWs. Overall, I am missing the article's contribution to policy and practice.

Response:

In the results section we have added a new table (Table 4) that compares the three configurations of supervision and location, with the intention that this will make the differences and similarities more explicit. (page 29 line 444)

We revised the Discussion section substantially to address the points you have raised. **Firstly**, we have expanded the comparison of our findings with those of other studies, and drawn out the policy implications for South Africa, by contrasting our findings with those of Marcus et al (page 32, lines 490-503).

“CHW supervision is often no more than periodic interactions by facility staff or officials who may not understand the context and role of CHWs, resulting in bureaucratic inspection and fault-finding, rather than problem solving and support.[6, 7, 20] For example, in the Ethiopian national programme, supervisors make irregular visits from either the local health centre or district offices.[21] In our study the presence of an on-site supervisor enabled on-going supportive engagement, rather than a distant supervisor, who had fit supervisory visits into a busy schedule. Evidence from a rapid assessment of the South Africa WBOT programme in five provinces suggested the facility-based staff were unable to provide effective supervision as they were overstretched, working in under-resourced and under-performing clinics. [15] The authors concluded that CHW teams should be based in stand-alone health posts with a nurse supervisor, rather than a clinic. Our data suggests it would be more advantageous for the nurse supervisor and the CHWs to be based the clinic, with clear rules about how much time the supervisor can spend on clinic activities.”

An explanation for how we will take forward this work in the intervention phase of the research is described on page 33 line 513-516.

“Training and supervision of the supervisors is a more neglected subject than that of CHWs.[9, 25, 26] This issue is beyond the scope of this paper, but in the second phase of our 3-year study, we are evaluating the effect on performance of a senior supervisor in building the capacity of the junior supervisors.”

A new paragraph (page 33-35 line 518-531) presents policy implications.

“There are on-going global efforts to scale-up CHW programmes as a means to strengthen health systems and accelerate universal health coverage. These include one million CHWs in sub-Saharan Africa,[27] making CHWs available everywhere on the African continent, [28] developing guidelines by WHO, as well as the latest funding commitment from the private sector.[29] Our study suggests that it is worthwhile to make senior supervision an explicit component of CHW programmes. It takes funds,

understanding of context specificity, as well as innovation, to implement cost-effective supervisory mechanisms. This research has offered some recommendations based on the experience of well-functioning teams (See Table 5)."

Please see next page for Table 5.

Table 5: Programme design features and supervisory strategies that worked

	Recommendation	Rationale
Programme design	● Attachment to PHC clinic	Facilitate physical & operational integration into the health system
	● Team up senior and junior supervisors	Build relations, guide CHWs to navigate through the community and system, pass down know-how to junior supervisors,
	● Setting guidelines as to how much time supervisors can spend assisting in the clinic	Help to acknowledge that the engagement is a two way collaboration, in which there are benefits for everybody
	● Strengthen HR management practices	Build trust, improve dialogue in the workplace, problem solving supervision, and culturally appropriate communication
Supervisory strategies	● Supervise home visits	Provide opportunities to strengthen CHWs' knowledge and skills, demonstrate a strong backup for CHWs in the community, keep updated of community's status
	● Formal and on-the-job training	Impart knowledge and skills
	● Regular debriefing/feedback	Individual and collective supervision, track performance, build up teamwork spirit
	● Examine daily logs and registers	Ensure accurate documentation and subsequent reporting
	● Direct CHWs to tasks, such as tracing defaulting patients, that explicitly	Ensure the clinic staff are able to see the benefits of having CHWs as part of the team

 ● assist the clinic Draw in clinic staff to work with and train CHWs 	Improve the extent and quality of working relationships between clinic staff and CHWs, and allows the CHWs to benefit from the clinic staff's expertise.
 ● Assist data collating & reporting 	Ensure that CHWs activities are accurately reported so that health system managers can see the benefits of the programme
 ● Administration and logistics 	Resolve administrative matters, negotiate for better work conditions, ensure CHWs are adequately equipped to deliver service

Reviewer: 3

Reviewer Name: Delanyo Dovlo

Institution and Country: WHO Regional Office for Africa, Congo

Please state any competing interests: None Declared

Comment:

The paper is well written and covers a subject of high importance –CHWs, an area that is again reemerging in global health as an instrument towards UHC. The CHW issue has remained rather ill-defined due to the fact that communities and their health issues differ widely and by nature, a CHW system has to be quite context related. Supervision is of course an essential element of whether CHWs function well or not – other important factors include training, equipment, incentives and motivators and the community acceptance (influenced by various other social and demographic characteristics).

Response:

Thank you for your comments. Yes, we agree that many factors determine the success of a CHW programme, and that they may affect one another. Our paper shows that supportive supervision can be motivating, supplement training and facilitate community acceptance. We have answered each of your points below.

Comment:

The paper focuses on supervision in a specific location of a province in South Africa. While the study environment and context is fairly well described, it may be difficult for people unfamiliar with the health systems in South Africa or the Gauteng Province to appreciate the levels of facilities involved and what types of nurses performed the supervision and these perhaps need to be elaborated on a bit more (define the difference between a health post and clinic, and between a senior nurse and junior nurses indicating a bit more on the difference in training. The training program afforded to the community health workers was fairly well described, but I could not find the training and preparation that the supervisors, whether senior or junior, had to undertake in order to play their roles. For example – will a junior nurse (enrolled nurse) with 20 years' experience, have better supervisory skills than the relatively junior ones compared with retired senior nurses in the study?

Response:

1. Regarding difference between a health and clinic, a new paragraph has been added from lines 74-87 on pages 7-8, to explain the role of the health post.

“Sedibeng District has established 16 health posts to provide community-orientated care, rather than the disease focused care provided by 28 primary health care clinics.[19] Some health posts were placed at some distance from a clinic, in order to improve access to basic services. A health post consists of one or two temporary structures (providing 3-6 rooms), without electricity and often with irregular water supply. It is managed by one or two professional nurses, who obtain medication and other resources via a ‘mother’ clinic. The nurses supervise the CHW team, and provide basic services such as chronic medication, immunisation and treating minor ailments.

At the time of the study there were 39 CHW teams (each with between 6-20 CHWs) in 37 of the district’s 72 wards (smallest geopolitical area). Sixteen of the teams were based at a health post and the remaining 23 were clinic-based. In addition to the services outlined in the national training programme (see above), the CHWs delivered medication to elderly or disabled chronically ill patients.”

2. Regarding the difference between the junior and senior supervisors and the preparation for their roles, the text has been expanded and now reads as follows (lines 204-220 on pages 15-16):

“The supervisors in our study sites were either senior (ie. professional) or junior (ie. Enrolled) nurses. A professional nurse (PN), with a 4-year degree in nursing, is able to diagnose patients, prescribe treatment and dispense medication. More importantly, the PN supervisors in the study site were also trained in primary health care and community nursing, and had attended various other courses on TB, HIV, diabetes and hypertension, integrated management of childhood illnesses (IMCI), nursing management and leadership. Many were rehired retirees, who had more than 30 years of experience as a nurse before they joined the programme (Table 3). Enrolled nurses (EN) have completed a 2 year course and can provide nursing care under supervision of a PN. All ENs except one (site 3B) received 1-week induction training on the WBOT programme before joining. Some of the EN supervisors in the site had also attended courses on early childhood development (ECD), TB, prevention of mother to child transmission (PMCT), and the expanded immunisation programme (EPI). Most of the ENs were younger and less experienced in community work than the CHWs they supervised. The PN supervisors reported the facility manager, as did the EN supervisors, if there was no PN supervisor in the team.”

Comment:

Supervision that is effective can be quite resource intensive – time, pay, travel costs etc). The senior nurses were rehired retirees, the junior nurses seemed to be younger in-service staff – were the junior nurses also solely dedicated to CHW supervision or had other facility duties? This may need to be more clearly explained – as the comparison with senior nurses supervisory advantage may be due to other factors – time, experience, community acceptance (of young supervisors for example).

Response:

We have added a new section to explain the clinic duties of both supervisors, explaining that the senior nurse’s supervisory advantage was not due to time, rather experience and acceptance by the CHWs. (page 21, lines 297-307)

There was an understanding between district officials and facility staff that clinic-based supervisors were to spend 70% of their time on CHW duties, and 30% of their time they could assist clinic staff, particularly if the clinic was short staffed. District officials regularly restated this 'rule' as they were concerned about supervisors neglecting their CHW duties. We did observe EN supervisors, who didn't have PN supervisors, spending the majority of their time in the clinic, but this was because they felt unable to contribute to CHW activities. However, one EN supervisor with PN mentor said:

"if I am not busy I help with vital signs stations, it is not part of my job but I do it anyway .. I refer my patients and the nurses see them. So there is teamwork. The support system is great so I never felt pressure to leave my WBOT duties" (EN interviews, 1B)"

Comment:

The effectiveness of community health workers and their supervisors and indeed the support from the facilities, I believe will involve alignment of a number of "hard" and "soft" incentives and motivations – some of which came out well in the study (eg building relations and being seen to support the pay and work issues of the CHWs). However, achieving work effectiveness for both CHWs and supervisors goes beyond their interrelationship and this could have been explored a bit more. For example; What advantage does CHWs bring to the clinics and health posts' work? How does these help facility staff to do their work better? Are there motivational factors for the retired senior nurses that the junior nurses lacked beyond being younger and inexperienced? It may be good also to explain to readers why rehired retirees were used and not current facility staff.

Response:

Thank you for this comment. In the findings section headed 'Collaboration between CHWs and facility staff', we have now expanded on this collaboration, explaining that there are benefits on both sides. We also explain how in the clinic-based teams there was a virtuous spiral with better supervision, leading to greater motivation and better performance, greater respect from the clinic staff, and greater collaboration (pages 24-27, lines 361-420):

Located at a distance from the main clinics, health post teams received little supervisory support from clinic staff. Other than supplying medication, there was little managerial oversight by the clinic. *"I am not directly supervising them except for sometimes when they are there. You can only supervise something that you can see. With the CHW team it is just via the phone and then it ends there."* (Facility Manager, 2A) In the health posts located in communities without a clinic, the PN spent a considerable amount of time seeing patients, and so neglected her CHW supervision responsibilities. Furthermore, the distance between the health post and the clinic hampered efforts to ensure CHW's patients accessed to care at the clinic.

"The saddest part is when the patient comes to the clinic and the file is not there, the doctor does not see the patient, and the patient is getting upset with us. Every time when we go to the clinic, we have to look for the files, the clerks do not assist us and we don't even find those files."(Observation notes from conversation with CHW, 2A)

In clinic based team, without a senior supervisor to assist, management of files was also a problem:

" Sometimes you find the admin clerk is in a bad mood, she speaks to us in a bad way in front of the patients, "I am not going to give you the file. You have to go back to the patient to tell the patient to come to the clinic to collect it." You report to the sister, she will tell you that she has been warning her for several times but she does not listen." (CHW-FGD, 3A)

CHWs reported that other facility staff expressed a dismissive attitude towards them:

"the peer educators, HIV/AIDS counselor we all go together to sign the same contract but they are treated as if they are more educated than us, they call us street maids" (CHW-FGD, 3B)

When asked why they were not allowed to work inside the facility, the CHWs responded: *"the manager tells us that we are not part of the clinic, so there's nothing she can do for us."* (field notes, 3A) The shortage of space exacerbated the CHWs' sense of marginalisation and exclusion.

However, in clinic based teams, with the support of senior supervisors, and the resulting growth in skills and confidence, CHWs had been able to establish better relationships with clinic staff. In turn, clinic staff, seeing the CHWs' greater motivation and effort, were more likely to treat the CHWs with respect. There were several benefits to this resulting collaboration. Firstly, communication and coordination was better. Working together, the facility manager and PN were able to confirm whether, once defaulters had been traced by CHWs, they returned to the clinic. *"We engage with facility manager, she helps with problems that we have... we include her in everything" (EN, 1B)*. Clinic staff willingly accepted referrals from CHWs, and included the CHWs in training sessions: *"Sister X is not (in the CHW team), but she is very helpful. She attended severe malnutrition training and she called all the CHWs and did a presentation for everybody." (EN, 1B)*.

Secondly, in one clinic, the contribution of CHWs had become indispensable: *"If they (CHWs) were not there, it means you will be over worked, the clinic will always be full, to be honest. There is good communication between us."* (Facility Manager, 1A)

The third benefit was the empowerment of CHWs, who learned to negotiate with other less willing staff: *"I have a patient who doesn't want to come to the clinic and I don't want my patient to default on her HIV treatment. I said to the sister 'my patient is working at the farms so please pack medication for her. I'll ask her to come next month.' The sister scolded me because the patient did not come on her last appointment. So I begged the sister and promised to bring the patient myself. Then it was sorted, the patient got her medication and she came to the clinic with me." (CHW-FGD, 1B)*

In the discussion we take this line of argument further, explaining how

(lines 450- 472 on pages 30-31)

“..we explored the effect of three different supervision and location configurations on CHW motivation and performance. Our findings show that experienced supervisors employed a range of strategies to train, motivate, monitor CHWs, to improve the quality of their work. In turn CHWs were treated with respect by the clinic staff, making space for collaboration and support that benefited both parties. Being an equal of the facility manager, a PN supervisor was in a position to bridge the divide between the CHWs and clinic staff. By obtaining a place for the CHWs inside the facility, as one of the supervisors did, she secured a position – physical and symbolic – for them in the health system. She drew in other staff to participate in the supervision of CHWs, further defending their legitimacy. Clinic staff provided advice and ensured the CHWs’ patients accessed the care that they needed; CHWs traced the defaulting patients in the clinics and so improved the clinics patient statistics. The senior supervisor, a shared asset, was able to effectively integrate the CHWs into the health system.

In contrast, the teams in health posts found it difficult to build relationships with the clinic, particularly if the health post is at some distance from the clinic and the PN was busy with patient consultations. The EN-headed teams lacked adequate training and supervision (partly due to their rural location), and consequently the CHWs’ performance was minimal. With little evidence of the CHWs’ contribution, the social distance between the CHWs and clinic staff, and her junior position, the EN was not able to facilitate a collaboration. Despite being based in the clinic, the CHWs were not effectively integrated into the health system.

A footnote (on page 37) has been added explaining why re-hired retirees were used.

“Re-hired retirees were used for three reasons. Firstly due to the shortage of professional nurses in the system, drawing in those who had retired, increased the supply. Secondly, they had a wealth of experience in community nurses, which younger nurses often did not. Thirdly, they are given an increment on top of their pension, and so cost the district less than a formally employed professional nurse.”

Comment:

The study described and discusses a number of interesting arrangements and interactions that produced a good effect (and some that did not) and does give quite a number areas to deduce ways of setting out a good supervisory mechanism for CHWs (indeed for all field workers) – about note taking, weekly statistics reviews; sampling previously visited households for the supervisor to visit; proving space to integrate CHWs work into clinic life etc etc., which could perhaps be summarized into a set of clearer recommendations of what worked and what didn't work. Arguably the sample and the context is limiting but can still generate enough dimensions for future reviews and studies.

Response:

Thank you for this useful suggestion. Table 5 has been added on page 35 to summarize our recommendations as follows.

Table 5: Programme design features and supervisory strategies that worked

Recommendation	Rationale
----------------	-----------

Programme design	● Attachment to PHC clinic	Facilitate physical & operational integration into the health system
	● Team up senior and junior supervisors	Build relations, guide CHWs to navigate through the community and system, pass down know-how to junior supervisors,
	● Setting guidelines as to how much time supervisors can spend assisting in the clinic	Help to acknowledge that the engagement is a two way collaboration, in which there are benefits for everybody
	● Strengthen HR management practices	Build trust, improve dialogue in the workplace, problem solving supervision, and culturally appropriate communication.
Supervisory strategies	● Supervise home visits	Provide opportunities to strengthen CHWs' knowledge and skills, demonstrate a strong backup for CHWs in the community, keep updated of community's status
	● Formal and on-the-job training	Impart knowledge and skills
	● Regular debriefing/feedback	Individual and collective supervision, track performance, build up teamwork spirit
	● Examine daily logs and registers	Ensure accurate documentation and subsequent reporting
	● Direct CHWs to tasks, such as tracing defaulting patients, that explicitly assist the clinic	Ensure the clinic staff are able to see the benefits of having CHWs as part of the team
	● Draw in clinic staff to work with and train CHWs	Improve the extent and quality of working relationships between clinic staff and CHWs, and allows the CHWs to benefit from the clinic staff's expertise.
	● Assist data collating & reporting	Ensure that CHW activities are accurately reported so that health system managers can see the benefits of the programme
	● Administration and logistics	Resolve administrative matters, negotiate for better work conditions, ensure CHWs are adequately equipped to deliver service

Comment:

With Africa's high NCD burden, it is good to see the NCD component of CHW work highlighted (rather than the usual focus on the global and donor supported initiatives. Perhaps a future study could look at the role of these CHWs in the non-communicable diseases challenge in communities.

Response:

Indeed, a future study could focus on CHWs in the management of NCDs. To highlight this feature of the CHW programme in South Africa, a brief paragraph has been added in Lines 26-30, page 5 in the Introduction section.

“Given the double burden of non-communicable and infectious diseases in LMICs, governments such as South Africa are seeking to shift CHW initiatives from focusing on a single disease or population group, to a more comprehensive approach. In 2011 South Africa has started to implement a national CHW programme (WBOTs; ward-based outreach teams) under a policy to strengthen primary health care.”

Comment:

I believe there should be some overarching oversight of supervisors, which has not been described in the study (who supervises them and how is that done? Does it influence their performance? Who do they report to?)

Response:

We have added the following sentence to lines 218-220 on page 16.

“The PN supervisors reported to the facility manager, as did the EN supervisors, if there was no PN supervisor in the team.”

Comment:

What is the policy within which the CHWs function? A brief summary will help an external reader.

Response:

A description of the policy is added in Introduction and Method.

In **Introduction**, (lines 28-32 page 5)

South Africa is implementing a national CHW programme (WBOTs; ward-based outreach teams) under a policy to strengthen primary health care [17]. The intention is to provide full-time on-site supervisors whose responsibility is to manage, train, mentor, monitor CHWs and to facilitate links with the health system and the community.

In **Method**, (lines 55-65, pages 6-7)

In some South African provinces, CHWs, previously employed by non-governmental organizations (NGOs), have been absorbed into a government programme (called WBOTS; ward-based outreach teams) in an attempt to shift from a patchwork of services to a programme with a national scope.[15, 18] The CHWs' role is predominately health promotion, prevention, screening and referral during household visits. Standardized CHW training covers identification of the need for ante-natal and post-natal care, monitoring immunisation of under 5s, adherence among patients with chronic diseases, screening for malnutrition, TB, gendered based violence, and making resulting referrals to health, social and other services. WBOT teams are meant to be comprised of a professional nurse, six or more CHWs, one health promoter and one environmental officer.[17]

Comment:

In general, I recommend these minor revision and clarification of information to make the article even more informative and relevant.

Response:

Thank you for the kind recommendation. We hope we have addressed your comments thoroughly.

VERSION 2 – REVIEW

REVIEWER	Karen Daniels South African Medical Research Council, South Africa
REVIEW RETURNED	23-Jul-2018

GENERAL COMMENTS	This is a much improved manuscript, but it does require one last good proof read.
---

REVIEWER	Abimbola Ayodele Olaniran Liverpool School of Tropical Medicine, United Kingdom
REVIEW RETURNED	28-Jul-2018

GENERAL COMMENTS	I commend the authors for identifying an important topic which is of great relevance to health system strengthening. However, they may require the services of a professional scientific writer in revising the manuscript to make the key messages clear and concise. - The reviewer also provided a marked copy with additional comments. Please contact the publisher for full details.
--

REVIEWER	Delanyo Dovlo WHO, Congo-Brazzaville
REVIEW RETURNED	20-Jul-2018

GENERAL COMMENTS	I have completed the second review and found a much improved and clearer paper that clearly links the research questions and methods to the results that were obtained. In particular, the limitations are now much clearer and describe the boundaries and expectations from the results.. I think the issue of supervision of CHWs is a very important one and while this is largely preliminary and covers a limited scope in terms of area covered (context being usually very critical), it does start to describe options for integrating CHWs through various supervisory systems. I also think it raises examples and method for supervisory motivation that can become guides for training and establishing supervisory systems for CHWs elsewhere. Though the methods may not be easily replicated due to its qualitative and context specific nature, it provides an acceptable basis for framing other studies with attributes to compare.
---

VERSION 2 – AUTHOR RESPONSE

Reviewers' Comments to Author:

Reviewer: 1
Reviewer Name: Karen Daniels
This is a much improved manuscript, but it does require one last good proof read.

Response:
Thank you. The suggestion is well received and a thorough proof read has been done.

Reviewer: 2
Reviewer Name: Abimbola Ayodele Olaniran
See file attached

Comment A01: (in Abstract-Introduction) The readers would benefit from knowing whether this is an observational study or the authors are referring to participant observation. Additionally, this sentence may be better placed in method section especially if “observation section” refers to a method of data collection.

Response:
As the reviewer suggested, we have specified use of qualitative methods in method section of the abstract as follows:

“In the initial observation phase of a 3-year intervention study, we employed multiple qualitative methods to study six CHW teams...” (page 2)

Comment A02: Unclear sentence. “Effective integration into the health system requires government financing with a national training programme, and integration of the CHW scope of work, supervision, referral networks and supply chain into the hierarchy of the health system.” (page 4, line 13-15)

Response:
This sentence has been rephrased as follows:

“Effective integration of CHW programmes into the health system requires government financing, national level planning, training and CHW scope of work, supervision, referral networks, and supply chains to be connected to and incorporated into similar processes provided for other cadres.” (lines 13-16, page 4)

Comment A03: I am missing an objective that looks into influence of integration and supervision on performance. The objectives of this study seem to focus on the processes without exploring the link between the processes (integration & supervision) and the outcome (performance) as suggested by the title of the manuscript.

Response:
Thank you for identifying the missing point. We have brought lines 49-51 (in the first revision copy) up to fill the gap. Now it reads as follows;

“By asking these questions, we examine supervision practices, the team’s integration into the health system, and their effect on CHW motivation and performance.” (lines 41-43, page 6)

Comment A04: This is a description of the study setting and population. This may be better placed under the method section.

Response:
As suggested, this description is now placed and reorganized in the method section as follows;

“In the initial observation phase (Sept 2016- Feb 2017) of a 3-year intervention study in Sedibeng Health District, Gauteng Province we used a case study approach to examine the operation of CHW teams with different configurations of supervisors and locations: (1) clinic-based teams supervised by a senior nurse (professional nurse; PN) and a junior nurse (enrolled nurse; EN); (2) community (health post)-based teams supervised by a PN and an EN; and, (3) clinic-based teams supervised by an EN only. We studied six teams (two of each of the three configurations); each team and their supervisors being a single case study.” (lines 46-53, page 6)

Comment A05: (regarding Results: CHW conditions of work across districts) The authors seem to veer off from the intended objectives of exploring the role of CHW integration and supervision in improving CHW performance. While some of these findings might still be relevant, it is important to situate them within findings that address the objectives. An example might be “CHWs without a high school diploma expressed the need for more supervisory support to provide services.

Response:

We agree with this comment. To build the relevance to our study objectives, a topic sentence is added in this paragraph to highlight systems problems which demotivated the CHWs.

“The lack of integration led to work conditions that were demotivating for CHWs.” (line 178, page 14)

Comment A06: (regarding Results: CHW conditions of work across districts) It is important to link these statements to the study objectives. Similar to the comment above.

Response:

In this paragraph, we added two sentences in the beginning and end, respectively.

“CHW work was hampered by insufficient provision of logistical support due to the lack of integration.” (lines 189-190, page 14)

“The general lack of resources compromised CHWs’ work and generated resentment.” (lines 195-196, page 15)

And in the following paragraph, separated from the previous, we highlighted the intervention of a supervisors and the improvement it brought about.

“Supervisors did try to address these deficiencies, and some of the endeavours appeared to be morale boosting.” (lines 200-201, page 15)

Comment A07: (regarding Results: Supervised home visits) Are these some of the factors influencing the effectiveness of supervision?

Response:

Indeed, they are. Proactive supervisors made a difference in such circumstances when and where they imparted knowledge and skills on the spot. On the other hand, we described unsatisfactory performance of CHWs to show that a less active supervisor failed to fulfil her supervisory role, and eventually failed CHWs.

In “The consequences of supervision” (starting from page 22), we presented the influence of inadequate training, formal and informal, on the ability of CHWs to assist patients. It has been summarized in Table 4.

I commend the authors for identifying an important topic which is of great relevance to health system strengthening. However, they may require the services of a professional scientific writer in revising the manuscript to make the key messages clear and concise.

Response:

Thank you. The manuscript has been carefully proofread. Reviewer’s edits in the text are much appreciated and have been accepted in the revised manuscript.

Reviewer: 3
 Reviewer Name: Delanyo Dovlo

I have completed the second review and found a much improved and clearer paper that clearly links the research questions and methods to the results that were obtained. In particular, the limitations are now much clearer and describe the boundaries and expectations from the results.

I think the issue of supervision of CHWs is a very important one and while this is largely preliminary and covers a limited scope in terms of area covered (context being usually very critical), it does start to describe options for integrating CHWs through various supervisory systems. I also think it raises examples and method for supervisory motivation that can become guides for training and establishing supervisory systems for CHWs elsewhere.

Though the methods may not be easily replicated due to its qualitative and context specific nature, it provides an acceptable basis for framing other studies with attributes to compare.

Response:

Thank you. The ongoing social experiment in one district of South Africa may not be replicated elsewhere, but its lessons can contribute to future deliberations of CHW programmes in other parts of South Africa and countries in similar settings. Soon we will be able to report further evidence using quantitative data to substantiate findings in this exploratory research.

VERSION 3 – REVIEW

REVIEWER	Abimbola Olaniran London School of Hygiene & Tropical Medicine
REVIEW RETURNED	01-Nov-2018

GENERAL COMMENTS	Once again, may I commend the authors for examining “the role of different levels of supervision and location of the CHW team on the team’s motivation, performance and integration into the health system”. I consider it an important study in view of the relevance of supervision to the success of CHW programmes. However, the authors would benefit from reviewing the recently launched WHO guideline on health policy and system support to optimize community health worker programmes (http://apps.who.int/iris/bitstream/handle/10665/275474/9789241550369-eng.pdf?ua=1). Subsequently, they would be able to discuss their findings in relation to the recommendations in the guideline.
--

VERSION 3 – AUTHOR RESPONSE

Reviewer: 2
 Reviewer Name: Abimbola Olaniran
 Institution and Country: London School of Hygiene & Tropical Medicine
 Please state any competing interests or state ‘None declared’: None declared

Once again, may I commend the authors for examining “the role of different levels of supervision and location of the CHW team on the team’s motivation, performance and integration into the health system”. I consider it an important study in view of the relevance of supervision to the success of CHW programmes.

However, the authors would benefit from reviewing the recently launched WHO guideline on health policy and system support to optimize community health worker programmes

(<http://apps.who.int/iris/bitstream/handle/10665/275474/9789241550369-eng.pdf?ua=1>).

Subsequently, they would be able to discuss their findings in relation to the recommendations in the guideline.

Response:

Thank you for the comment and suggestion. In response to the recommendations in the WHO guideline, we have revised the last two paragraphs of the discussion section (pages 33-34, lines 528-542), which now reads:

Globally, there are on-going efforts to scale-up CHW programmes as a means to strengthen health systems and accelerate universal health coverage. These include the call for one million CHWs in sub-Saharan Africa,[27] making CHWs available everywhere on the African continent, [28] new funding commitments from the private sector,[29] as well as the recent WHO guideline on health policy and system support to optimize CHW programmes.[30]

The new WHO guideline states that *“planners should adopt ‘whole-of-system’ approaches, taking into account health system capacities and framing the CHW role vis-à-vis other health workers in order to appropriately integrate the CHW into the health system”*. [30] It also indicates that while there is benefit of supportive supervision, there is a lack of granular evidence. Our study provides rich evidence on the range of supervision strategies, how the effectiveness of these varies by the level of supervisor and whether the team has its base in a facility or not. Taking a whole-system perspective has enabled us to draw conclusions about the barriers and facilitators to integrating the CHW into the system. Below we offer recommendations based on the experience of well-functioning teams to complement current global efforts (See Table 5).